# Your Actions Talk: DUET – A Multimodal Dataset for Contextualizable Dyadic Activities

## Abstract

Human activity recognition (HAR) has advanced significantly with the availability of diverse datasets, yet the field remains limited by a scarcity of datasets focused on two-person, or "dyadic," interactions. Existing datasets primarily cater to single-person activities, overlooking the complex dynamics and contextual dependencies present in interactions between two individuals. Failing to extend HAR to dyadic settings limits opportunities to advance areas like collaborative learning, healthcare, robotics, augmented reality, and psychological assessments, which require an understanding of interpersonal dynamics. To address this gap, we introduce the Dyadic User Engagement dataseT (DUET), a comprehensive dataset designed to enhance the understanding and recognition of dyadic activities. DUET comprises 14,400 video samples across 12 interaction classes, capturing the highest sample-to-class ratio of dyadic datasets known to date. Each sample is recorded using RGB, depth, infrared, and 3D skeleton joints, ensuring a robust dataset for multimodal analysis. Critically, DUET features a taxonomization of interactions based on five fundamental communication functions: emblems, illustrators, affect displays, regulators, and adaptors. This classification, rooted in psychology, supports dyadic human activity contextualization by extracting the embedded semantics of bodily movements. Data collection was conducted at three locations using a novel technique that captures interactions from multiple views with a single camera, thereby improving model resilience against background noise and view variations. We benchmark six state-of-the-art, open-source HAR algorithms on DUET, demonstrating the dataset's complexity and current HAR models' limitations in recognizing dyadic interactions. Our results highlight the need for further research into multimodal and context-aware HAR for dyadic interactions, and provide a dataset to support this advancement. DUET is publicly available at "Anonymized DUET Repository", providing a valuable resource for the research community to advance HAR in dyadic settings.

## 1 Introduction

### 1.1 Motivation

Human activity recognition (HAR) is a field within artificial intelligence focused on identifying and analyzing human actions from sensor data, and it has achieved significant success across various domains. The success of HAR can be attributed to many factors, including the commitment of the field to producing publicly available datasets that can be used to help refine data-driven deep learning algorithms across various contexts. While there is an abundance of HAR datasets already available, the majority pertain to single-person—or *monadic*—activities. A better understanding of two-person—or *dyadic*—interactions is essential for enhancing the accuracy, responsiveness, and overall capabilities of systems where human interaction plays a central role.

Dyadic interactions, which involve the interplay between two individuals, convey deeper communicative and cultural significance. Despite their complexity, HAR for dyadic interactions offers several advantages. The inclusion of a second subject improves the performance of many HAR tasks by introducing an additional distinguishing factor (Adeli et al., 2020). For instance, consider the actions "waving in" and "thumbs up." These two movements appear similar at first glance, as both

involve extending one's arm. However, their small-scale hand movements differ only slightly, making them difficult to distinguish in isolation. The distinction becomes much clearer when another subject is involved—specifically, by observing the initiating action of one subject and the reaction of the other. The reacting subject may physically approach if the initiating subject waves them in, while they may simply nod in acknowledgment if the initiating subject gives a "thumbs up". These differing responses provide valuable contextual cues that improve the accuracy of recognizing and differentiating between the two activities. The study of dyadic interactions allows for a more accurate understanding of human behaviors that are absent in monadic activities, enabling systems to better interpret and respond to social dynamics. For example, telepresence avatars in augmented reality provide digital representations of participants during remote conferences. By analyzing the subtleties of interactions between individuals, dyadic activity analysis enhances user experience and increases the authenticity of digital environments (Ahuja et al., 2019). Similarly, social robots designed to provide companionship for children leverage dyadic analysis to recognize dangerous situations and intervene in a timely, adaptive manner. These robots also utilize two-person datasets to deliver more natural and engaging conversational interactions, supporting the social, cognitive, and emotional development of children (Chen et al., 2022). Additionally, in public infrastructure, accurately recognizing dyadic social activities enhances safety by detecting potential dangers and enables the provision of more personalized services in public spaces (Coppola et al., 2020).

## 1.2 Review of existing datasets

Despite these advantages, the availability of dyadic datasets remains limited, particularly in comparison to the abundance of monadic datasets. This scarcity poses an increasing challenge as interactions between humans and technical systems grow more complex. The research community's uneven emphasis on these two activities types is reflected in their differing recognition performance levels. Lin et al. (2024b) showed that monadic algorithms, which have achieved outstanding benchmarking records for monadic activities, do not perform nearly as well for dyadic interactions. This highlights the disparity between monadic and dyadic activities, which stems from the greater variety of expressive and cultural signals, as well as the increased complexity of spatial and temporal coordination between two or more subjects. To reconcile this discrepancy and improve dyadic HAR, there is a need for more datasets tailored to dyadic interactions. As highlighted in the IEEE Control Systems Society's report on control for societal-scale challenges, traditional boundaries between humans and technology are blurring, and emerging fields like cyber-physical-human systems (CPHS) face challenges in designing robust interactions between humans and control systems (Annaswamy et al., 2023). One of the central CPHS research challenges is characterizing how humans adapt during interactions. Dyadic datasets are critical for developing models that can enhance system adaptability, safety, and trustworthiness in these complex environments (Annaswamy et al., 2023).

Besides increasing the number, diversity, and quality of dyadic datasets, contextualizing activities has proven effective in improving the performance of HAR tasks (Niemann et al., 2021). Contextualization distills meanings embedded in body language, such as emotional and cultural significance, adding another layer of comprehension to the tracking of bodily movements. For instance, a "thumbs up" signifies approval in most Western cultures but represents a profanity in Greece and several Middle Eastern countries. Contextualization enables the interpretation of the cultural significance of gestures, such as recognizing the nuanced meanings of a "thumbs up." In addition to enhancing HAR accuracy, contextualization supports the development of various downstream applications. For example, certain branches of CPHS investigate how humans interact with and benefit from the built environment (Doctorarastoo et al., 2023a;b). A critical aspect of this framework is understanding the embedded semantics of human behaviors through bodily movements. This understanding provides stakeholders with deeper insights into system use, improving infrastructure design, maintenance, and operation. Contextualization also paves the way for automating psychological and sociological assessments—such as sociometric tests (Moreno, 1941)—that currently rely on self-reported data. These manual evaluations are labor-intensive and also prone to attribution bias. By integrating contextualization with dyadic HAR, these processes can be automated, extracting user preferences from bodily movements (Lin et al., 2024a) and addressing these limitations. For instance, contextualization enhances telepresence avatars by capturing nonverbal cues and paralinguistic signals, improving the quality and authenticity of remote communication (Ahuja et al., 2019).

Despite these recognized benefits, the few available dyadic datasets—listed in Table 1—are inadequate for extracting the underlying semantics of bodily movements. While some datasets focus

on healthcare activities, others are restricted to tracking bodily movements within specific action categories. No existing dataset selects activity classes using scientifically grounded methods that prioritize semantic cohesion to capture the social embeddings of activities. This lack of structured selection limits the ability to understand functional relationships between actions, hindering models from generalizing effectively to new, unlabeled behaviors. A dyadic dataset that fully supports contextualization is still absent in the research community.

### 1.3 OBJECTIVES AND NOVELTY OF THIS PAPER

To enhance HAR performance for dyadic activities through contextualization, we introduce the Dyadic User Engagement dataseT (DUET). Featuring 12 taxonomized interactions, DUET helps to bridge monadic and dyadic HAR while connecting HAR to other disciplines. It is publicly available under an MIT License at "Anonymized DUET Repository" (Authors, 2024).

Instead of repeating previous approaches that arbitrarily select activity categories, our dataset is built on a psychology-based classification that identifies five core communication functions in human interactions: emblems, illustrators, affect displays, regulators, and adaptors. This taxonomy provides a scientifically grounded framework for integrating HAR with interdisciplinary applications. For instance, lie detection often relies on emblematic slips—unconscious, fragmented gestures that deceivers attempt to suppress while lying. Similarly, emotion detection heavily depends on adaptors, which reveal physical or emotional discomfort. By capturing interactions from all five categories, DUET addresses critical gaps left by existing datasets. This stands in stark contrast to existing datasets, as shown in Table 1, particularly the largest dyadic dataset to date, NTU RGB+D 120 (Liu et al., 2019). While NTU RGB+D 120 includes dyadic interactions, it represents only three of the five categories—illustrators, affect displays, and regulators. This imbalance prevents it from supporting applications that require a comprehensive understanding of the taxonomy. In contrast, DUET deliberately incorporates interactions from all five categories, ensuring semantic cohesion and preserving the functional relationships between actions. This design enables HAR to generalize more effectively, recognizing both labeled and unlabeled actions by aligning them with shared traits of existing categories. As a result, DUET facilitates connections between HAR and fields like psychology, sociology, and behavioral sciences, paving the way for applications like automated emotion recognition and the analysis of social behaviors in complex, real-world scenarios. By bridging this gap, DUET stands as a critical step toward advancing both HAR and its wider applications.

The dataset was collected using the Microsoft Azure Kinect v2 (Microsoft, 2024a) (hereafter referred to as the Azure Kinect), a high-quality, multimodal camera capable of capturing RGB, infrared (IR), depth, and 3D skeletal joint data. Over the span of one year, 23 participants contributed to the dataset, generating 14,400 video samples, with 1,200 samples recorded for each interaction category. To our knowledge, this dataset features the highest sample-to-class ratio published to date.

The testbeds consist of three locations across a university campus in the United States (US): an open indoor space, a confined indoor space, and an outdoor area. These settings were chosen to represent a range of environments where human activities commonly take place. For example, the companionship and support provided by social robots may take place in a small bedroom (confined indoor space). A sociological evaluation of a group of students' social connectivity during class could be conducted in a large auditorium (open indoor space). A potential application for CPHS is to redesign public open spaces based on patterns of measured usage and socialization to foster user sociability and cohesion (open outdoor space). This variety not only allows downstream applications to leverage DUET for investigating the direct and indirect impacts of ambient surroundings on algorithm performance but also improves the resilience of deep learning models against background noise. We intentionally collected data at various times and on different days to capture a range of environmental conditions (e.g., lighting) and background noise, ensuring the dataset reflects real-world scenarios. Another challenge is the limited number of views, which can affect system robustness (Perera et al., 2020). The lack of multiple views in existing literature (Table 1) undermines the generalizability and view-invariance of video samples from different orientations. To address this, we propose a novel data collection process that captures interactions from multiple angles using a single camera. This low-cost approach captures activities from various orientations, something that even multiple cameras have struggled to achieve. We evaluate the performance of six open-source, state-of-the-art algorithms using RGB, depth, and 3D skeleton joint data. This comparison not only highlights the complexity of contextualizable dyadic interactions but also reveals the strengths of each modality.

Table 1: comparison of existing dyadic datasets shows that the proposed dataset has the *highest number of samples per class*, the most views, and a relatively high number of locations. Note: (1) "Views" refer to different sensor orientations from which interactions are captured, and (2) "background noise" indicates the presence of random people's movement or cluttered environments.

| Dataset | Modalities | #Videos | #Classes | #Locations | #Views | Background noise | Year |
|---|---|---|---|---|---|---|---|
| UT Interaction (Ryoo et al., 2010) | RGB | 160 | 6 | 2 | 1 | No | 2010 |
| SBU Kinect (Yun et al., 2012) | RGB+D+J | 300 | 8 | 1 | 1 | No | 2012 |
| JPL Interaction (Ryoo & Matthies, 2013) | RGB | 399 | 7 | 5 | 1 | No | 2013 |
| G3Di (Bloom et al., 2016) | RGB+D+J | 168 | 14 | 1 | 1 | No | 2015 |
| $M^2I$ (Liu et al., 2018) | RGB+D+J | 1,760 | 9 | 1 | 2 | No | 2015 |
| ShakeFive 2 (Van Gemeren et al., 2016) | RGB+J | 153 | 8 | 1 | 1 | No | 2016 |
| PKU-MMD (Liu et al., 2017) | RGB+D+J+IR | 4225 | 10 | 1 | 3 | No | 2017 |
| MMAct (Kong et al., 2019) | RGB+keypoints+ acceleration+ orientation+ Wi-FI+Pressure | 2162 | 2 | 4 | 4+ego | No | 2019 |
| NTU RGB+D 120 (Liu et al., 2019) | RGB+D+J+IR | 24,828 | 26 | - | 155 | No | 2019 |
| Air Act2Act (Ko et al., 2021) | RGB+D+J | 5,000 | 10 | 2 | 3 | No | 2020 |
| **DUET (our dataset)** | **RGB+D+J+IR** | **14,400** | **12** | **3** | **360** | **No** | **2024** |

The remainder of the paper is structured as follows. Section 2 details the taxonomy for classifying human interactions. Section 3 overviews the dataset, including modalities, format, acquisition configurations, biometrics, annotation, and data splits for cross-location and cross-subject evaluations. Section 4 benchmarks six open-source algorithms and their results. Finally, Section 5 presents conclusions, key takeaways, and future directions.

## 2 CONTEXTUALIZING HUMAN INTERACTIONS

A social interaction is an exchange of information between two or more individuals, and the delivery can happen through various channels. Among all communication channels, bodily movement represents a critical part of social interaction as instinctive actions convey unspoken cues of the conversation or communication (Sharan et al., 2022). To study the context embedded in social interactions through bodily movement, generating a dataset that attempts to exhaust all existing interactions would not be feasible. Similarly, selecting actions arbitrarily would detract from the dataset's ability to preserve functional relationships and shared characteristics among actions, which is needed to create semantic cohesion across classes. In this work, we propose a dataset, DUET, in which the selection of interactions is not arbitrary but instead grounded in psychological principles.

A total of 12 kinesic interactions are drawn from a taxonomy developed by Ekman & Friesen (1969), which classifies human interactions into five groups based on their fundamental communication functions. This system provides a structured and effective approach for categorizing interactions and systematically extracting the information embedded in bodily movements. The categories include emblems, illustrators, affect displays, regulators, and adaptors:

- Emblems: Emblems are gestures that have direct verbal translation and can be culturally specific. The same gesture might be interpreted differently for different cultures (Hartman, 2024). For instance, a "thumbs up" indicates well done in most Western cultures, but is a derogatory sign in Middle Eastern countries. Interactions chosen are *"waving in," "thumbs-up,"* and *"hand waving."*

- Illustrators: Bodily movements that illustrate the verbal message they accompany are called illustrators, which are used to clarify conversations and are context dependent (Chute et al., 2023). Interactions chosen are *"pointing"* and *"showing measurements."*

- Affect displays: Affect displays are gestures that reveal one's affective and emotional state. An example of an affect display is "arm crossing," which can signal defensiveness, insecurity, or anxiety. Interactions chosen are *"hugging," "laughing,"* and *"arm crossing."*

- Regulators: During interactions, regulators determine the alternation of instigating and receiving. Interactions chosen are *"nodding," "writing circles in the air,"* and *"holding one's palms out."* Nodding is a gesture of acceptance and acknowledgement used for the continuation of the conversation. "Drawing circles in the air" displays the need to expedite the conversation. "Holding one's palms out" is used to warn the other person to cease the conversation.

- Adaptors: Adaptors are habitual movements that satisfy personal needs and can be used to increase or decrease emotional stability (Neff et al., 2011). The interaction chosen is *"twirling or scratching hair"* to moderate one's stress during contemplation.

## 3 DATA COLLECTION AND MANAGEMENT

### 3.1 DATA MODALITIES AND DATA FORMAT

For the data collection, we use the high-quality and multimodal Azure Kinect, equipped with an RGB camera, a depth sensor, and an IR sensor. These sensors all operate at 30 frames per second for three seconds for each video sample, yielding 91 frames per sample. The recorded data is saved in the Matroska ('.mkv') container format, allowing multiple tracks of data formats to be extracted through post-processing. Tracks of modalities used in this dataset are RGB, depth, IR, and 3D skeleton joint sequences.

The specification of each data format varies depending on the conventions commonly used in the research community: each RGB frame is captured with a resolution of $1{,}920 \times 1{,}080$ and is stored in a '.jpeg' format. We record depth and IR sequences with a resolution of $640 \times 576$ and store them as 24-bit '.png' files. The skeleton joints of every sample video are stored in their corresponding '.csv' files. Each file contains a $91 \times 193$ array, where each row represents a frame, and each column holds information related to that frame. The first column records the timestamp of the frame, and the following 96 columns capture the $x$, $y$, and $z$ coordinates of 32 joints of one subject (as illustrated in Figure 2a), measured as the distance (in millimeters) from the joint to the camera. For instance, the first three columns record the $x$, $y$, and $z$ values of the first joint. The order of the joints follows the joint index in (Microsoft, 2024b). The last 96 columns record the 32 joints of the other object.

Figure 1 presents sample frames from each action category across different modalities, each offering distinct strengths and weaknesses. RGB frames capture rich details such as interactions, locations, and subject features, making them highly informative but lacking in user privacy protection. However, since RGB frames compress the 3D world into a 2D plane, they often suffer from issues like occlusion and view variation. In contrast, 3D skeleton joints provide the spatial position of each joint in a 3D space, offering a desirable view-invariant characteristic. Beyond joint positioning, 3D skeletons reveal little about the subject's identity, making this modality more privacy-friendly. This privacy feature is particularly valuable in human-centered applications such as smart homes, CPHS, and elder care management. Overall, the comparison of these modalities highlights an inverse relationship between privacy and the amount of information conveyed—the more information a modality provides, the less it typically protects user privacy. Our dataset includes four modalities that span this entire spectrum, encouraging both the exploration of individual modalities and the fusion of multiple modalities to balance privacy preservation with information richness.

### 3.2 DATA ACQUISITION AND SETUP

After selecting the Azure Kinect as the sensing device, a setup for housing the sensor was needed to guarantee consistency throughout the experiments. We constructed a sensing module, shown in Figure 2b, which positions the Azure Kinect 215 cm above the ground and tilts it forward at a 37° angle. This setup allows for capturing interactions with a full field of view and minimal occlusions.

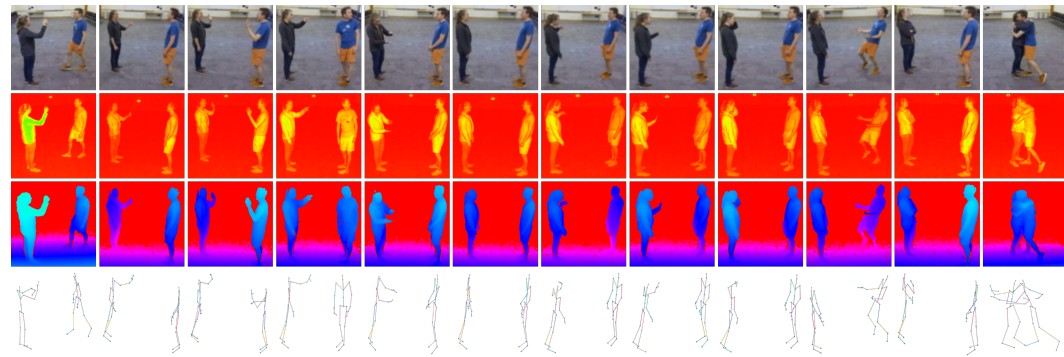

Figure 1: Sample data from 12 interactions. The modalities are, from top row to bottom row: RGB, IR, depth, and 3D skeleton joints. The 12 interactions are, from left to right: "waving in," "thumbs up," "waving," "pointing," "showing measurements," "nodding," "drawing circles in the air," "holding one's palms out," "twirling or scratching hair," "laughing," "arm crossing," and "hugging."

An important aspect of the experiment is the selection of testbed locations. Rather than attempting to cover all possible environments, we chose three representative locations across a US university campus: an open indoor area, a confined indoor space, and an outdoor area, as shown in Figure 3. These locations are selected to provide a variety of backgrounds and support the exploration of the effects of the ambient environment on the sensors. A common limitation of HAR datasets is the lack of diverse backgrounds, which can lead to deep learning models overfitting to background noise. By conducting the experiment in three distinct locations, we aim to improve the generalizability of background noise handling. We also acknowledge that a contextualizable dataset should be applicable across a range of environments, such as parks, schools, nursing facilities, and smart homes. Collecting data in varied locations, especially outdoors, allows for the examination of how the ambient environment directly and indirectly affects sensor performance and algorithm accuracy.

Since the experiment was conducted at three different locations, it was essential to ensure the data collection process was consistent and repeatable. To achieve this, we designed a testbed setup, shown in Figure 2c, which was used across all three environments. In this setup, volunteers were asked to perform each interaction 40 times within a rectangular area marked on the ground. After each repetition, a beep signaled the participants to rotate either clockwise or counterclockwise before proceeding to the next repetition. This structured process helped minimize labeling ambiguity by ensuring that subjects performed each action in a predefined sequence, one action at a time. This approach allowed us to confidently associate specific images with their corresponding actions, effectively eliminating the potential for ambiguity or labeling errors. In less controlled settings, where actions may overlap or occur simultaneously, we recommend incorporating contextual tags to enhance label clarity and reduce ambiguity in the data.

The benefits of this innovative technique are two-fold. First, it enabled us to capture interactions from a wide range of orientations relative to the camera. As shown in Figures 6 and 7 in Section A, some frames capture the side profiles of the subjects, while in others, one subject faces the camera while the other has their back to it. This diversity in orientations enhances the view-invariance of HAR algorithms. Second, our dataset includes samples with occlusions—a common challenge in HAR tasks. Occlusion occurs when one subject fully or partially obstructs the other within the camera's field of view. By incorporating occlusions, our dataset aims to help HAR algorithms address this issue more effectively. Furthermore, capturing multiple viewpoints using a single camera reduces deployment costs, as achieving similar results would otherwise require multiple sensors. Although the environments for this dataset were curated, similarly to other datasets in Table 1, we intentionally collected data at different times of the day and on various days to capture a wide range of environmental conditions. For example, in the outdoor setting, some participants performed during the early morning or late afternoon when the lighting was dim, while others were assigned to midday sessions under bright sunlight. On several occasions, the sky was fully overcast, providing a low-light environment. These variations in illumination are evident in Figures 6 and 7. In the indoor environments, we enriched the lighting conditions by opening curtains to allow natural light to filter in and by configuring the overhead lights differently for each session. Additionally, the outdoor setting introduced further variability, including breezes and higher winds that caused rustling in the

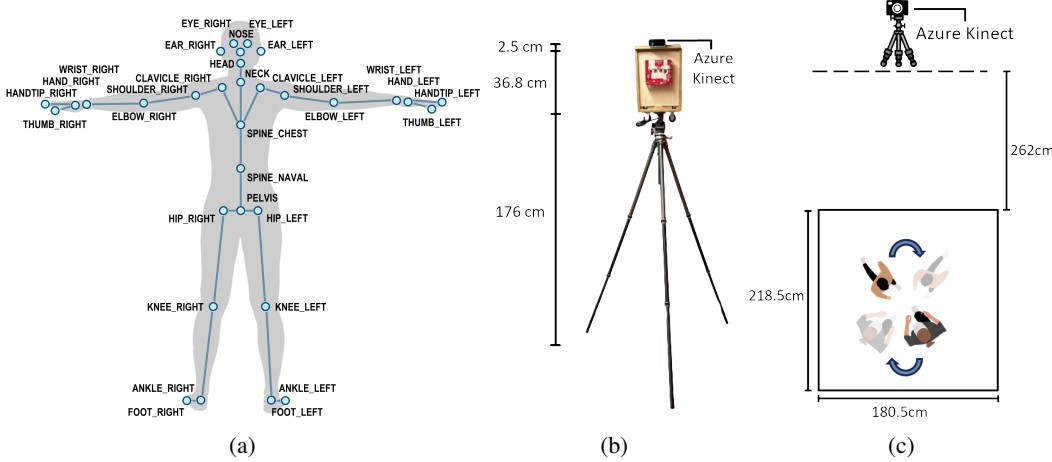

(a)                (b)               (c)

Figure 2: Using the Azure Kinect SDK (Microsoft, 2023), (a) 32 3D skeleton joints are extracted following this labeling scheme. (b) The sensing module configuration and (c) bird's-eye view of the testbed remain consistent across locations, with subjects confined to a rectangular area.

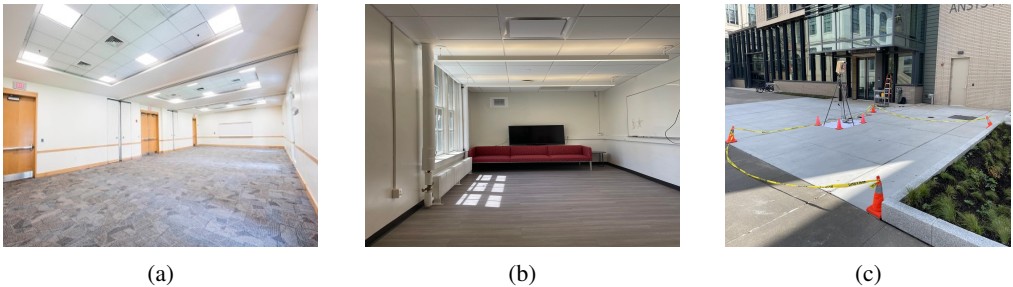

(a)                (b)               (c)

Figure 3: Testing locations: (a) open indoor area, (b) confined indoor space, and (c) outdoor area.

surrounding trees, creating varying levels of background noise. An active construction site located behind the testing area also contributed to the diversity of conditions, with noticeable changes in the site's layout and equipment placement from one test to another. By incorporating these variations in lighting, noise, and environmental dynamics, DUET more closely mirrors real-world scenarios, enhancing its relevance and robustness for human activity recognition tasks.

### 3.3 SUBJECTS

A total of 15 male and eight female subjects participated in the experiments. The subjects were randomly paired to perform actions across the three locations. The subjects' ages range from 23 to 42 years old with a mean age of 27 years and standard deviation of 4.01 years. Heights ranged from 165.1 cm to 185.4 cm with a mean height of 172.7 cm and a standard deviation of 8.46 cm. The subjects' weights ranged from 55 kg to 93 kg with a mean weight of 69 kg and a standard deviation of 10.1 kg. To further enhance the diversity and robustness of the dataset, users are encouraged to apply data augmentation techniques to create additional variations and improve the generalizability of machine learning models using this dataset.

### 3.4 DATA ANNOTATION

To simplify the file compilation, we organized the data into a folder structure, as shown in Figure 4. The folder structure comprises four hierarchical layers: (1) modality, (2) location combination, interaction label, and subject, (3) timestamps, and (4) image or '.csv' files. The first layer classifies files by modality, including RGB, depth, IR, and 3D skeleton joints. The next layer uses a six-digit code, *LLIISS*, to categorize the location, interaction label, and subject. In this code, *LL* represents the location: *CM* for the indoor open space, *CC* for the indoor confined space, and *CL* for the outdoor space. *II* refers to the numbered activities (1–12) listed in Table 2, and *SS* indicates the subject pair, ranging from 1–10. Note that the same subject pair number in different locations does not indicate

Table 2: Activity labels and their corresponding interactions.

| Lable ID | Dyadic interaction | Label ID | Dyadic interaction |
|---|---|---|---|
| 1 | Waving in | 7 | Drawing circles in the air |
| 2 | Thumbs up | 8 | Holding one's palms out |
| 3 | Waving | 9 | Twirling or scratching hair |
| 4 | Pointing | 10 | Laughing |
| 5 | Showing measurements | 11 | Arm crossing |
| 6 | Nodding | 12 | Hugging |

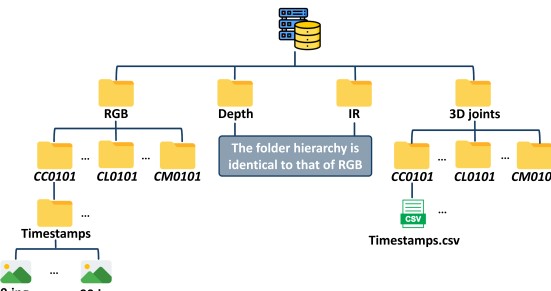

Figure 4: The data folder structure for our dataset is designed to ensure easy access for users. The RGB, depth, and IR modalities follow the same hierarchical structure, while the 3D skeleton joint folders store all 3D coordinates for a sample video clip in a single '.csv' file.

the same pair; only the pairs *CCII02* and *CLII07*, *CCII01* and *CMII10*, and *CCII03* and *CMII05* represent the same individuals across locations. As mentioned earlier, each pair was asked to repeat an interaction 40 times, and all repetitions were recorded in a single video. To segment the video temporally, we organized each time window by start and end timestamps. For example, a folder named 40800222_43800211 contains a recording that begins at 40800222 and ends at 43800211 milliseconds after the Azure Kinect is connected. Inside each timestamp folder, the corresponding clip is stored frame by frame, with frames numbered sequentially from 0–90.

### 3.5 CROSS-LOCATION AND CROSS-SUBJECT EVALUATIONS

One of the key motivations for creating DUET is to encourage the research community to explore HAR in the context of dyadic, contextualizable interactions. To support this, we provide a baseline training and test data split for evaluating algorithm performance. In addition to the standard cross-subject evaluation, we also include a cross-location evaluation. We recognize that applications involving dyadic, contextualizable interactions may take place in a variety of indoor and outdoor settings, so the cross-location evaluation helps ensure HAR algorithms are resilient to location variation. For the cross-subject evaluation, we use *CCII05*, *CCII07*, *CLII01*, *CLII05*, *CMII06*, and *CMII09* for the test data, and the remainder for the training data. For cross-location evaluation, *CCIISS* is selected as the test data, while *CLIISS* and *CMIISS* are used as the training data.

### 4 BENCHMARKING STATE-OF-THE-ART HAR ALGORITHMS

In this section, we evaluate the performance of six open-source, state-of-the-art HAR models with publicly available code, as listed in Table 3. This work intentionally selects algorithms that are open-source to ensure that the implementation used in our benchmarking is consistent with the original benchmarking conducted by the algorithm's developers. This decision prioritizes reproducibility and transparency, both of which are essential for meaningful comparisons. Since DUET provides multiple modalities, the evaluation includes two RGB-based, two depth-based, and two skeleton-based algorithms. The results of the evaluation are presented in Table 3.

First, we analyze the effect of occlusion on the RGB modality, as its accuracy is relatively lower compared to other modalities. As previously mentioned, occlusion is a common challenge in vision-based HAR. To evaluate its impact, we train the two selected algorithms using only unoccluded

Table 3: Cross-location and cross-subject accuracy comparison for RGB, depth, and 3D skeleton joints. Note: the parenthesized values are accuracies for unoccluded samples.

| HAR algorithm | Modality | Cross-location accuracy ( % ) | Cross-subject accuracy ( % ) |
|---|---|---|---|
| DB-LSTM (He et al., 2021) | RGB | 9.65 (13.81) | 17.85 (21.34) |
| V4D (Zhang et al., 2020) | RGB | 8.26 (18.58) | 7.79 (34.68) |
| DOGV-ST3D (Xiaopeng et al., 2021) | Depth | 13.15 | 18.77 |
| DB-LSTM (He et al., 2021) | Depth | 14.94 | 23.18 |
| PAM-STGCN (Yang et al., 2020) | 3D joints | 30.73 | 36.65 |
| DR-GCN (Zhu et al., 2021) | 3D joints | 38.17 | 41.57 |

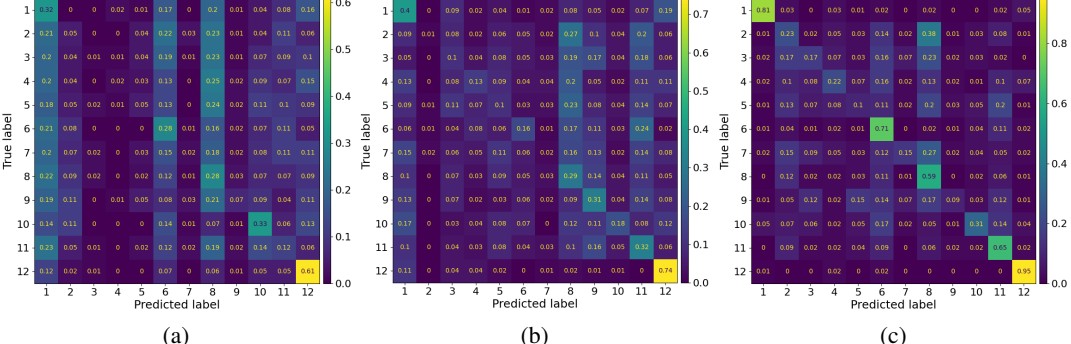

(a)          (b)          (c)

Figure 5: Representative confusion matrices for cross-subject evaluation for (a) RGB (DB-LSTM (He et al., 2021)), (b) depth (DB-LSTM (He et al., 2021)), and (c) 3D skeleton joints (DR-GCN (Zhu et al., 2021)). Note: each label's interaction corresponds to the mapping in Table 2.

samples for both cross-location and cross-subject evaluations. The corresponding results (Table 3) are comparable to the benchmarking records of other datasets (Liu et al., 2017). The results indicate that both algorithms show improved performance when occluded samples are excluded. This experiment highlights not only the significant impact of occlusion on algorithm performance but also the critical importance of including occluded samples in datasets for comprehensive evaluation.

Overall, the cross-subject evaluation outperforms the cross-location evaluation across all modalities in the state-of-the-art algorithms, which can be explained by two key factors. First, RGB-based and depth-based algorithms are prone to learning view-dependent motion patterns, often correlating background with motion trajectories during training. In the cross-subject evaluation, the training set includes samples from three locations, whereas in the cross-location evaluation, only two locations are used for training. As a result, these models struggle to generalize to unseen backgrounds during testing, leading to lower accuracy in the cross-location evaluation. Second, the difference in the number of training samples also contributes to the performance gap. In the cross-subject evaluation, 80% of the dataset—approximately 11,520 samples—is used for training, while in the cross-location evaluation, only two-thirds of the dataset is available for training. Performance improves with a larger training sample size. These two phenomena are also present Liu et al. (2019)'s work.

Another observation is the gradual increase in accuracy of the state-of-the-art HAR algorithms tested in our study, progressing from RGB to depth, and then to 3D skeleton joints, which aligns with the expansion of dimensional information. RGB-based algorithms compress input into a 2D plane, leading to lower accuracy since human interactions involve both 3D spatial and temporal coordination (Lee & Kim, 2022). This dimensional compression limits the system's ability to fully capture spatial dynamics. Adding depth information to each pixel in an image, as seen in depth-based algorithms, provides an additional layer of information. The improvement in performance with depth inputs is particularly clear when we compare the same model (i.e., DB-LSTM) using RGB and depth inputs separately. However, despite the increase in accuracy from RGB to depth modalities, both still leave room for improvement. This is due to the fact that both modalities operate in Euclidean space (i.e., images), making them more susceptible to view variations. DUET addresses this issue and improves accuracy by providing more robust data. Additionally, training in Euclidean space can be easily in-

fluenced by trivial features. As shown in Fig. 5a and Fig. 5b, RGB and depth models are confused by common poses shared across activities—for example, standing is present in nearly all activities. In contrast, skeleton-based algorithms perform HAR in non-Euclidean space (Peng et al., 2021), representing human interactions in 3D space relative to the camera, leading to better accuracy.

Skeleton-based algorithms outperform other modalities because they capture activities in a 3D space relative to the camera, which is well-suited for the spatial complexity of human interactions. These algorithms can extract underlying motion patterns regardless of the viewpoint. Additionally, 3D skeletons provide a sparse representation of the human body, which helps prevent the network from learning irrelevant features. However, this sparsity can also hinder recognition in certain cases. In our dataset, many dyadic interactions differ only in subtle ways. For example, both the "thumbs up" gesture and "holding one's palms out" (i.e., label ID 2 and 8, respectively) involve arm extension, but the former requires raising the thumb, while the latter involves holding the hand vertically. The simplified skeletal representation may not capture these fine distinctions using current HAR algorithms. This is evident in Figure 5c, which shows these two actions are frequently confused by the algorithm. While the nuances are more apparent in RGB and depth images, from which the 3D skeleton joints are extracted, state-of-the-art skeleton-based algorithms still struggle to detect them.

## 5 DISCUSSION AND CONCLUSION

In this work, we introduce DUET, a contextualizable dataset consisting of 12 dyadic interactions, based on a psychological taxonomy that organizes human interactions into five groups according to their communication functions. This taxonomy advances the field of HAR beyond simple body movement tracking by extracting the embedded semantics in dyadic interactions. Moreover, contextualizing human activities enhances HAR models and paves the way for significant downstream applications, such as autonomous vehicles, urban infrastructure planning, and healthcare.

14,400 samples were collected across 12 interactions, resulting in 1,200 samples per activity—the highest sample-to-class ratio published. The samples span four modalities: RGB, depth, IR, and 3D skeleton joints, each offering unique strengths. The multimodal dataset encourages both the individual use of each modality to refine models for specific applications and the fusion of modalities to combine complementary information, maximizing the value provided by each modality.

DUET also aims to improve the view and background invariance of HAR models. We introduce a novel data collection procedure to capture human interactions from multiple angles using a single camera, something previously unachievable even with multiple sensors. This innovative setup enhances resilience to variations in viewing angles, reflects real-life scenarios where observations are not restricted to a specific angle, and reduces deployment costs. The choice of testbed locations is carefully considered. Data was collected in three distinct environments: an open indoor area, a confined indoor space, and an outdoor area. This variety not only improves generalizability but also enables applications to assess how ambient environments affect system performance.

To establish baseline performance for DUET, we evaluate six HAR algorithms with open-source code to ensure an accurate assessment of their capabilities—two RGB-based, two depth-based, and two skeleton-based algorithms. While some previous work has attempted to recognize dyadic interactions using monadic algorithms, the performance reveals a persistent gap between recognizing monadic and dyadic activities. In this study, we take a step further by benchmarking six dyadic algorithms with our dataset. The results highlight (1) the complexity of social interactions that remains underexplored in existing literature, and (2) the vulnerability of HAR algorithms to changes in view and background, which presents new research opportunities for future investigation.

Future developments from this work can be broadly categorized into two areas: refining HAR algorithms and enhancing the taxonomy to better capture the embedded semantics of interactions. As shown in Table 3, all modalities require improvement when it comes to contextualizable dyadic interactions. In addition to developing more sophisticated HAR models capable of capturing the nuances in these interactions, another way to enhance performance is through contextualization. We have laid the groundwork for contextualizing human activities by integrating a psychological taxonomy with HAR. The next step is to further define this framework, mapping all interactions to their corresponding levels of embedded meaning, which can benefit downstream applications such as CPHS, autonomous vehicles, smart homes, and healthcare.

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

## A   APPENDIX

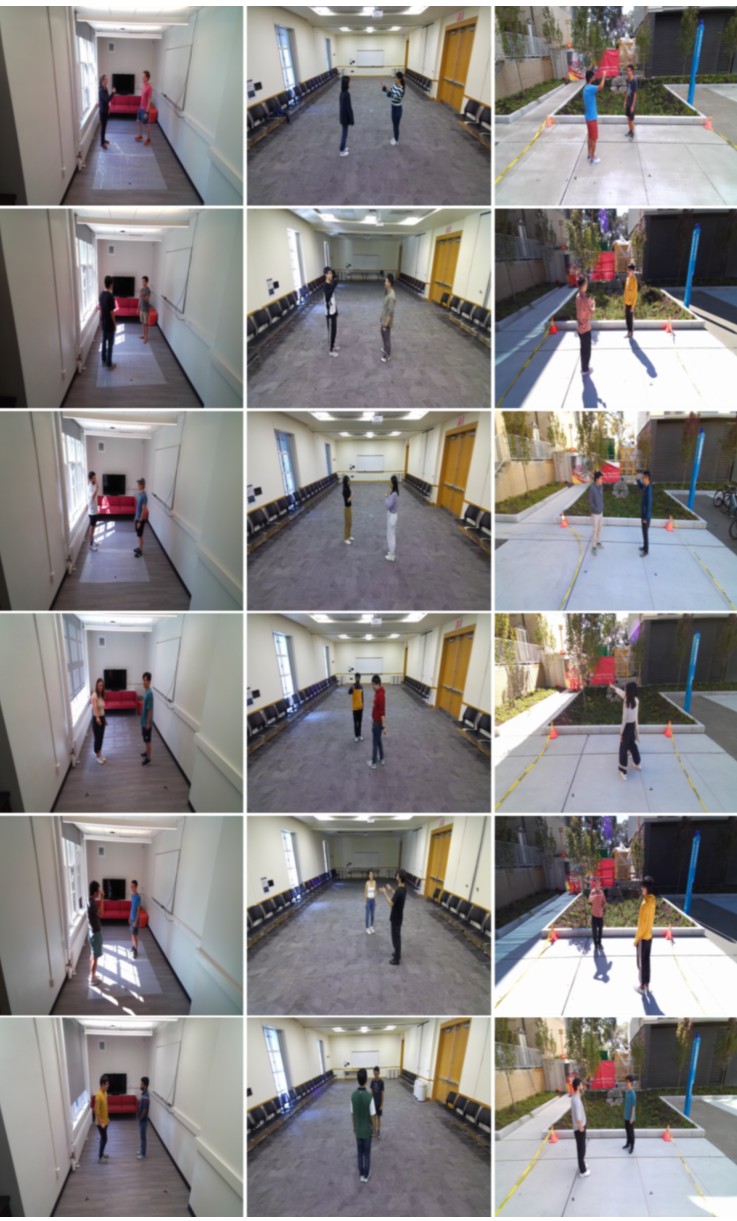

Figure 6: Sample data from the first six interactions. The locations presented are, from left to right: the confined indoor space, the open indoor space, and the open outdoor space. The six interactions are, from the top to bottom rows: "waving in," "thumbs up," "waving," "pointing," "showing measurements," and "nodding." These images demonstrate the variation in lighting conditions, viewpoints, and occlusions.

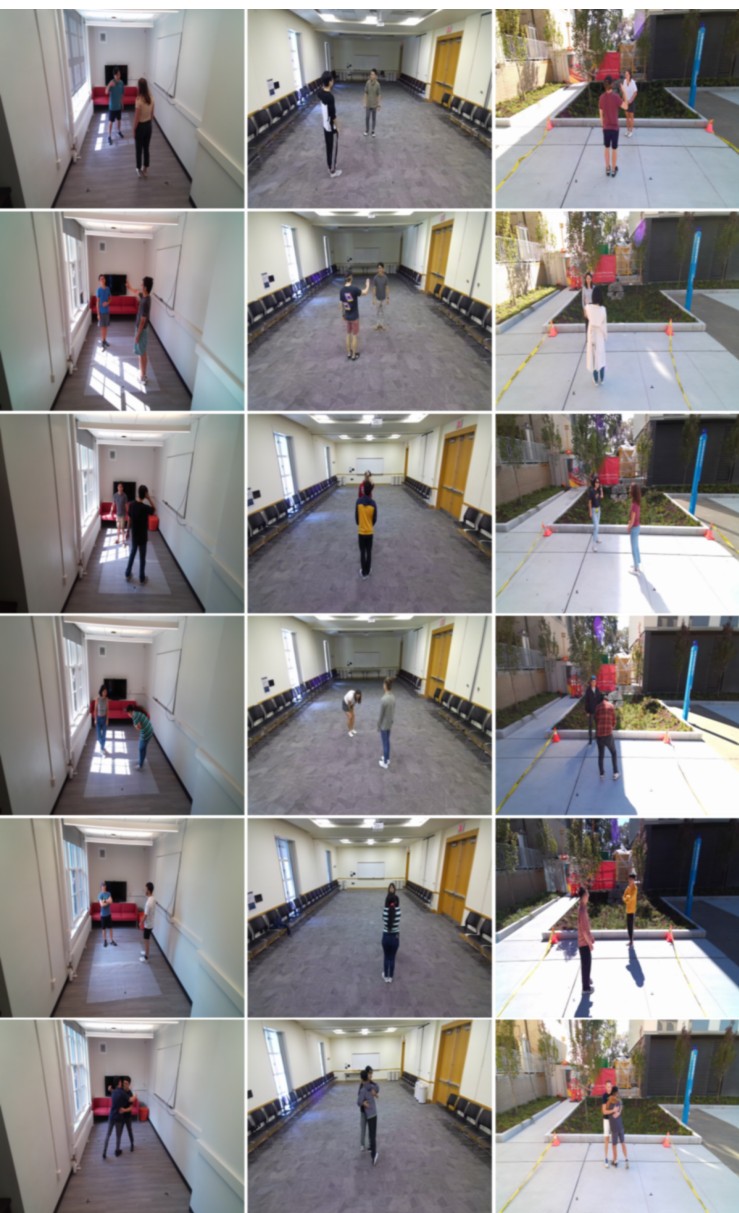

Figure 7: Sample data from the last six interactions. The locations presented are, from left to right: the confined indoor space, the open indoor space, and the open outdoor space. The six interactions are, from the top to bottom rows: "drawing circles in the air," "holding one's palms out," "twirling or scratching hair," "laughing," "arm crossing," and "hugging." These images demonstrate the variation in lighting conditions, viewpoints, and occlusions.

