# OpenReview forum: "Your Actions Talk: DUET - A Multimodal Dataset for Contextualizable Dyadic Activities"
_ICLR.cc/2025/Conference — Submitted to ICLR 2025_

### Official Review · Reviewer_1oSi · 2024-10-31

**Soundness:** 3
**Presentation:** 3
**Contribution:** 2
**Rating:** 6
**Confidence:** 4

**Summary:**

This work proposes the Dyadic User Engagement dataset (DUET), a comprehensive collection aimed at improving the understanding and recognition of dyadic activities. Notably, the authors introduce 12 kinesic interactions based on a taxonomy developed by Ekman and Friesen. DUET comprises 14,400 video samples across these 12 interaction classes, with each sample recorded using RGB, depth, infrared, and 3D skeleton joints. Data collection was conducted at 3 locations using a unique technique that captures interactions from multiple angles with a single camera. To demonstrate the dataset’s complexity and highlight the limitations of current human activity recognition (HAR) models in identifying dyadic interactions, the authors benchmark 6 existing open-source HAR algorithms on DUET. Overall, this work is intriguing, and its organization is well-structured.

**Strengths:**

The paper demonstrates a commendable level of originality by introducing the Dyadic User Engagement dataset (DUET). This dataset not only addresses a previously underexplored area of human activity recognition (HAR) but also incorporates a taxonomy of kinesic interactions. By combining multiple data modalities (RGB, depth, infrared, and 3D skeleton joints), the authors present a new dataset that enhances the understanding of dyadic interactions, making a significant contribution to the field.

The quality of the research is evident in the methodology employed for data collection and evaluation. The authors have outlined their experimental setup, including the selection of diverse locations and the pairing of subjects. Additionally, data collection was conducted at 3 locations using a unique technique that captures interactions from multiple angles with a single camera. The benchmarking of 6 existing HAR algorithms against DUET further validates the dataset's robustness and relevance.

The paper is well-structured and clearly communicates its objectives, methodology, and findings. The use of figures, such as those depicting the experimental setup aids in conveying ideas effectively.

The significance of this work lies in its potential impact on the field of human activity recognition and related applications, such as social interaction analysis and behavior recognition. By providing a comprehensive dataset that captures the nuances of dyadic interactions, the authors provide the way for future research to build on their findings.

**Weaknesses:**

Limitations of Controlled Environments
This study effectively utilizes diverse locations with subjects, however, the controlled nature of the experiments might not capture the complexity of real-world interactions. The reliance on a structured setup might limit the realism of the data collected. The authors should discuss the limitations of controlled environments in more depth. Providing examples of how real-world interactions differ from the study setup would strengthen this analysis. The paper should include detailed descriptions of how participants were instructed and how the experiments were designed to achieve 360 views.

**Questions:**

Comment 1: I understand that the authors are emphasizing the consistency in data quality by noting that all datasets have no background noise. However, I would like clarification on what "No" indicates in the Background and Noise column of Table 1. Does this mean that all datasets lack background noise, or does it indicate that this aspect has not been evaluated?

Comment 2: The visualized actions depicted in Figure 2 are unclear, particularly in the RGB and skeleton data. The figure should be illustrated more clearly to allow readers to observe it carefully.

Comment 3: The authors selected three representative locations across a US university campus: an open indoor space, a confined indoor space, and an outdoor area. This variety provides different backgrounds and supports the exploration of the effects of the ambient environment on the sensors. In the camera setup, participants were asked to perform each interaction 40 times, rotating either clockwise or counterclockwise before the next repetition. This technique captures interactions from a wide range of orientations relative to the camera, enhancing view invariance. However, the controlled actions in a controlled environment might not fully capture the complexity of real-world interactions, limiting realism. While rotating participants aids in achieving view invariance, biases may still arise based on their familiarity with the camera setup. Additionally, it would be helpful to provide more detailed descriptions of how participants were instructed or how the experiments were designed to achieve 360-degree views.

Comment 4: In Figure 3b, the illustration mentions that the distance between the Kinect camera and the experimental area, where two participants perform interactions, is 262 cm. The experimental area, a rectangular space marked on the ground, is 218.5 cm by 180.5 cm. Could you please provide reasoning for this setup?

Comment 5: Please provide more details about how subjects were randomly paired and the significance of this pairing for the study, as it would enhance understanding. For example, why is pairing important for creating the DUET dataset?

Comment 6: In the cross-location and cross-subject evaluations, the authors mention specific folders in the dataset (e.g., CCII05, CCIISS). However, please explain what these folders represent (e.g., location and subject pairs) and why these specific data sets were chosen for evaluations.

Comment 7: For performance metrics, the results are presented. If possible, please include concrete examples of how the evaluations are conducted or what specific challenges they address to help readers with visualization. For example, Cross-Subject Evaluation: In this scenario, a model is trained on hand waving interactions performed by one pair of subjects (Pair A) and then tested on a different pair (Pair B). The challenge is the variability in gestures, such as differences in grip strength, which tests the model’s ability to generalize and accurately identify the hand waving despite individual differences. Cross-Location Evaluation: Here, the same hand waving interaction is performed in two distinct settings: a large open room and a confined office space. The model is trained on data from the open room and tested in the office. The challenge lies in the differing backgrounds and spatial dynamics, assessing the model’s accuracy under varying environmental contexts.

**Details Of Ethics Concerns:**

The study involves human subjects, and it would be necessary to ensure informed consent and ethical treatment throughout the data collection process. Detailed information on how participants were recruited, informed about the study, and how their data will be used is essential. Please confirm these matter such as Privacy, security and safety/ Responsible research practice (e.g., human subjects, data release).

---

### Official Review · Reviewer_fiGq · 2024-11-03

**Soundness:** 2
**Presentation:** 2
**Contribution:** 2
**Rating:** 3
**Confidence:** 5

**Summary:**

The paper presents DUET, a new dataset for Human Activity Recognition for two-person interactions. DUET consists of 14,400 samples across 12 interaction classes, captured using four modalities: RGB, depth, infrared and 3D skeleton joints. The dataset includes data from both indoor and outdoor settings. DUET features a taxonomization of interactions based on psychologically motivated communication functions. The authors benchmark six state-of-the-art open-source HAR algorithms on DUET.

**Strengths:**

This is a large-scale dataset for two-person interactions involving subtle actions such as laughing, thumbs-up gesture etc. I am not aware of any similar dataset, so this could be useful for research in social interactions.
The paper is mostly well-written and easy to follow.

**Weaknesses:**

Limited technical contribution: The paper’s primary contribution is a dataset, and there is minimal technical innovation in dataset processing or model development, which makes it a weak submission for ICLR.
There is no clear validation that the taxonomization helps, or indeed if it is, in any way, related to the problem of activity recognition.
The actions include subtle movements such as laughing, thumbs-up, which are not exactly dyadic, and perhaps too subtle to be analyzed from wide-angle cameras capturing full body motions.
A set of algorithms have been evaluated on the dataset without proper explanation as to why they are ideally suited for this problem. The algorithms considered are somewhat dated, i.e., the recent models from last 2-3 years have not been considered.

**Questions:**

1. How can actions such as laughing and thumbs up be recognized using wide-angle captures from afar?
2. Why haven't some of the recent State of the Art methods for HAR included in the evaluation.

---

### Official Review · Reviewer_gyyk · 2024-11-04

**Soundness:** 3
**Presentation:** 2
**Contribution:** 2
**Rating:** 3
**Confidence:** 3

**Summary:**

This paper presents a new dataset, called DUET, focusing on dyadic human activities. The data consisted of 23 participants doing prescribed activity repetitions in 3 locations, resulting 14,400 RGB+D+J+IR videos. Authors asked participants to rotate during repetitions, and hence were able to capture activities from 360 different views while keeping the camera/sensor stationary.

Authors include evaluation of 2 RGB-based, 2 depth-based and 2 skeleton-joint-based algorithms from the literature as initial benchmarks.

**Strengths:**

* Authors tried to base their selection of activities/interactions classes in their dataset in psychological principles. Hence, the chosen activities are well thought-out.

* Authors tried to collect the activity data from diverse viewpoints, recognizing that research struggle to train deep learning algorithms that generalize easily to new viewpoints otherwise.

**Weaknesses:**

The weaknesses of this work are:

* There are only 23 participants contributing 14400 videos (each person giving ~600 videos). The number of participants is very small, given the size of today's ML models. Hence, I wonder if there is risk of models overfitting on the individuals on this datasets. Authors need to discuss this risk and address how/why this dataset is usable even when it consists of only 23 participants' data.


* It is not clear what the advantage of this dataset is over NTU RGBD 120 dataset (Liu et al., 2019), which has 106 participants, ~25000 videos, and 26 dyadic human activities. Even when NTU RGBD 120 dataset is five years old, it seems bigger that the proposed DUET dataset in all the above metrics. I think, author need to add more participants to either scale DUET to be bigger than NTU RGBD 120 dataset or discuss why DUET dataset is needed in addition to NTU RGBD 120 dataset.


* I am skeptical of authors' results in Table 3. The two state-of-the-art RGB-based activity detection algorithms seem to achieve only ~9% accuracy for 12-class classification problem. That accuracy is basically the accuracy of random-chance classifier on 12-class classification problem (1/12 = 8.5%). While I am not familiar with those two works, I am skeptical that modern RGB-based activity detection algorithms have the accuracy as bad as random chance.
    * I think, the authors need to provide sufficient details about these experiments so that a reader can confirm that authors were able to run DB-LSTM and V4D algorithms correctly. A good way to do that would be to include their numbers on other datasets (e.g. NTU RGBD 120 dataset) and compare them with DUET numbers.
    * Is the low accuracy because the number of locations and participants is too small for modern deep learning algorithms? (are we  essentially seeing overfitting on location and/or participants?).
    * Does the accuracy improve if you choose only a subset of classes?
    * If the low accuracy numbers are, in fact, correct, the author need to investigate and explain in detail why the accuracy so low.

**Questions:**

* In Fig. 6, authors only include label indices in the confusion matrix. What are the class names corresponding to those indices? Without those, a reader cannot interpret the confusion matrix.

---

### Official Review · Reviewer_Rkrq · 2024-11-04

**Soundness:** 3
**Presentation:** 3
**Contribution:** 2
**Rating:** 3
**Confidence:** 4

**Summary:**

This paper presents the Dyadic User Engagement dataseT (DUET), a comprehensive multimodal dataset specifically crafted to enhance the recognition and contextual understanding of dyadic (two-person) human interactions. DUET includes 14,400 video samples, divided into 12 activity classes based on a taxonomy by Ekman and Friesen that categorizes interactions according to fundamental communication functions (e.g., emblems, affect displays). The dataset is distinctive for its high sample-to-class and view-per-class ratios, achieved through recordings in three varied locations—open indoor, confined indoor, and outdoor settings—ensuring resilience to environmental variations. Each DUET sample captures data in four modalities: RGB, depth, infrared, and 3D skeleton joints, offering a detailed, privacy-conscious representation of interactions. The paper also benchmarks state-of-the-art human activity recognition (HAR) algorithms on DUET, revealing challenges in generalizing models to dyadic settings and emphasizing the need for further research in context-aware HAR. DUET, publicly available under an MIT license, provides a novel resource for advancing HAR in applications such as collaborative learning, healthcare, and robotics.

**Strengths:**

- The paper addresses the dyadic human activity recognition, filling a key gap with a well-structured dataset focused on complex, two-person interactions.

- The study applies a scientifically grounded methodology, categorizing activities using Ekman and Friesen’s taxonomy based on core communication functions (e.g., emblems, affect displays), enhancing interpretability in HAR tasks.

- DUET provides high-quality multimodal data (RGB, IR, depth, and 3D skeletons) and achieves the highest sample-to-view ratio, supporting robust, view-invariant HAR models.

- Data was collected across varied settings (open indoor, confined indoor, outdoor) to ensure resilience against background variations, enhancing model generalizability.

**Weaknesses:**

- NTU-RGB 120 is larger and with more interactive action classes, the author should discuss the features where the difference of the proposed dataset compared with the existing ones in a detail

- Lack of comparisons with the existed important dataset for interactive action understanding:

PKU-MMD: A Large Scale Benchmark for Continuous Multi-Modal Human Action Understanding

MMAct: A Large-Scale Dataset for Cross Modal Human Action Understanding

- Organization of the paper is hard to follow, the introduction section is long without subsection split by mixing the related work regarding the dataset part in it,  and no related work section to summarize the current HAR methods for interactive action understanding.

- With the proposed dataset, what is the new the community could benefit from it, or how the method design will be effected from it is not clear. Only a new dataset with the same domain problem set up with limited amount of new data which is challenge to say the significant contribution to the ICLR community level.

- Current benchmarking evaluates algorithms using only single modalities (e.g., RGB or depth) rather than combining them. Testing modality fusion (e.g., RGB with depth) could reveal if multimodal approaches improve accuracy.

- The dataset has a gender skew toward male subjects, which may limit model generalizability across diverse user groups. A more balanced representation would improve robustness.

- While DUET spans three settings (open indoor, confined indoor, outdoor), broader real-life scenarios (e.g., crowded or low-light environments) could enhance real-world applicability.

- Some interactions (e.g., "hugging" vs. "arm crossing") may be open to interpretation, introducing labeling ambiguity. Adding contextual tags could improve label clarity.

**Questions:**

Please see the weakness.

The current contribution is not significant enough as an ICLR paper.

**Details Of Ethics Concerns:**

As mentioned in the paper, the authors already have the consent from the participants (Line 149 to 150) for the release of the dataset. It should be fine for the privacy. However, the full dataset is not released due to double blind policy. Recommend authors to check videos carefully before releasing to the public.

---

> ### Comment · Reviewer_Rkrq · 2024-12-02
> **Thansk for the response from author**
>
> Thank you for the author's respones, regarding the novelity and comparision with existed dataset,
> the current new part described in the rebuttal is still very limited, if taxonomy is the one of the main contribution, the other similar dataset e,g., UTD-MHAD is needed to be compared, the atom level action design idea is covered in this dataset.
> Moreover, there is no experiment to support the author's dataset's new features mentiond in the novelity such as,
> no experiment to show how taxonomy task can be solved by using this dataset, and the complex, real-world scenarion action recognition performance can be benifit from this datset or any new method designed for this.
>
> I will keep my rating according to the current technical contribution limitation.

---

### Meta-Review · Area_Chair_E734 · 2024-12-08

**Metareview:**

This work proposes the Dyadic User Engagement dataset (DUET), a comprehensive collection aimed at improving the understanding and recognition of dyadic activities. Notably, the authors introduce 12 kinesic interactions based on a taxonomy developed by Ekman and Friesen. DUET comprises 14,400 video samples across these 12 interaction classes, with each sample recorded using RGB, depth, infrared, and 3D skeleton joints. Data collection was conducted at 3 locations using a unique technique that captures interactions from multiple angles with a single camera. To demonstrate the dataset’s complexity and highlight the limitations of current human activity recognition (HAR) models in identifying dyadic interactions, the authors benchmark 6 existing open-source HAR algorithms on DUET.

Advantages: The paper is well-organized. The significance of this work lies in its potential impact on the field of human activity recognition and related applications, such as social interaction analysis and behavior recognition.

Disadvantages:
- The paper lacks thorough comparisons with existing major datasets for interactive action understanding (e.g., NTU-RGBD 120, PKU-MMD, MMAct) to highlight its unique advantages or differences.

- The contribution of the new dataset to the community is unclear; it does not appear substantially larger, more diverse, or more challenging than existing datasets, raising questions about its overall significance.

- Current benchmarking focuses solely on single-modality evaluations; the potential benefit of multimodal fusion (e.g., RGB + depth) remains unexplored.

- The dataset is male-skewed, risking poor model generalization, and lacks sufficient participants for robust, large-scale training, which may lead to overfitting concerns.

- Broader environmental settings and clearer annotations are needed to enhance real-world applicability and reduce labeling ambiguity.

- The reported baseline results are suspiciously low, suggesting possible implementation issues or overfitting problems, and more experimental details or comparison with known benchmarks are required.

- The technical novelty is not significant, and the paper does not convincingly show how its taxonomy or dataset resolves existing challenges or advances the field, making it a weak submission for ICLR.

I think the current manuscript cannot meet the bar, though the idea behind this paper is good.

**Additional Comments On Reviewer Discussion:**

All the reviewers have some major concerns on this paper regarding the significance of the datasets, the unclear reasons for low-performing baseline methods, the male-skewed distribution of datasets, the limited environmental settings and annotations. The authors only explain the reasons but do not provide enough justifications via experiments or releasing more details. Hence the reviews are not convinced. Thus, I think the paper does not reach the ICLR acceptance bar.

---

### Decision · Program_Chairs · 2025-01-22

Reject